# Impact of preventive chemotherapy on transmission of soil-transmitted helminth infections in Pemba Island, United Republic of Tanzania, 1994–2021

**Shaali Ame[1], Fatma Kabole[2], Alphoncina Masako Nanai** [3], **Pauline Mwinzi[4],**
**Denise Mupfasoni** [5]**, Said Mohammed Ali[1], Antonio Montresor** [5]*

**1** Public Health Laboratory, Ivo de Carneri Foundation, Pemba, United Republic of Tanzania, **2** Neglected Tropical Disease Control Programme, Ministry of Health, Community Development, Social Welfare, Gender, Elderly and Children, Zanzibar, United Republic of Tanzania, **3** United Republic of Tanzania Country Office, World Health Organization, Dar es Salaam, United Republic of Tanzania, **4** Expanded Special Project for Elimination of Neglected Tropical Diseases, World Health Organization, Regional Office for Africa, Brazzaville, Congo, **5** Department of Control of Neglected Tropical Diseases, World Health Organization, Geneva, Switzerland

* montresora@who.int

**Data Availability Statement:** All relevant data are within the manuscript and its Supporting Information files.

## Abstract

Soil-transmitted helminth (STH) infections cause significant morbidity in children and women of reproductive age. The World Health Organization (WHO) recommends preventive chemotherapy (PC) of at-risk populations with anthelminthics to control these infections. Historically, STH are very intensively transmitted in Pemba Island (Zanzibar). A survey conducted in 1994 in 12 schools estimated a STH prevalence near to 100%. This extremely high prevalence induced the introduction of PC in the island; initially, however, PC was not regularly administered because of difficulties linked to drug procurement. A second STH survey, conducted in 2011, in 24 schools estimated a prevalence of STH of 89%; after this survey, PC was regularly administered until 2018. We conducted a survey in 2021 using the same method as that used in 2011. The prevalence of STH was evaluated at 80% (95% CI 78.1–81.5) and most of the STH cases were due to *Trichuris trichiura*. More than 32% (95% CI 30.3–34.0) of the children investigated had infections of moderate or heavy intensity. PC has been conducted for over 25 years in Pemba Island. However, despite its beneficial impact, both the prevalence and the intensity of STH infections remain high, and the intervention has been insufficient in controlling STH morbidity. This is probably due to a combination of irregular PC, climatic conditions favourable to STH transmission, the low sensitivity of *T. trichiura* to benzimidazoles, high population density and poor sanitation. Improvement of sanitation coverage remains a key measure to permanently reduce the prevalence and intensity of STH. Possible changes to the present PC approaches to better control STH in Pemba would be (i) to assure high coverage in all schools, (ii) to use mebendazole instead of albendazole given its better activity on *T. trichiura* and (iii) to use a combination of ivermectin and mebendazole to further increase anthelminthic efficacy on *T. trichiura*.

**Funding:** The author(s) received no specific funding for this work

**Competing interests:** The authors have declared that no competing interests exist.

## Author summary

A survey conducted in 1994 in school aged children showed that soil-transmitted helminth were very intensively transmitted in Pemba Island (Zanzibar) with prevalence near to 100%. Between 1994 and 2011 several rounds of administration of anthelminthic were organized and a second survey was conducted in 2011 showing a decrease of prevalence and intensity of infection.

The present paper reports the results of a third survey conducted in 2021 showing a further decline of the prevalence and a persistence of *T. trichiura* as the more prevalent parasite (in 1994 hookworms were the most transmitted STH). The interventions conducted until now where not sufficient to eliminate the morbidity caused by these parasites and it is suggested to include ivermectin in the anthelminthic distributed periodically.

## Background

The group of soil-transmitted helminths (STH) includes *Ascaris lumbricoides*, *Trichuris trichiura* and the two hookworm species *Ancylostoma duodenale* and *Necator americanus*. STH cause a relevant disease burden in tropical and subtropical countries [1] where levels of sanitation are inadequate. Current estimates indicate that over 1.5 billion people are infected with at least one of these helminth species [2]. STH infections cause severe morbidity that is proportional to the number of worms infecting the host [3]. Infections of moderate to heavy intensity (MHI) are considered to cause the greater part of STH morbidity [4].

In countries where STH are endemic, in addition to improvement in sanitation and behavioural change interventions, the World Health Organization (WHO) recommends preventive chemotherapy (PC), a periodic treatment with anthelminthic medicines, to different groups of populations at risk [5]. STH are historically known to be very intensively transmitted in Pemba Island (Zanzibar). A survey conducted in 1994 in 12 schools [6] estimated a STH prevalence near to 100%, with over 76% of individuals with MHI infections. A second STH survey conducted in 2011 in 24 schools [7] estimated a prevalence of STH of 89% (95% CI 88–91) with 29% (95% CI 27–30) of individuals infected at MHI.

Since 1994, PC activities have been conducted in the island. During 1994–2011, the records of PC were imprecise, but, generally, one or two rounds of deworming were distributed every year to schoolchildren depending on the availability and sufficiency of resources. Albendazole and ivermectin were also distributed annually to the entire population of Pemba Island in the context of the Global Programme to Eliminate Lymphatic Filariasis [8], which was implemented in Zanzibar by the Ministry of Health, Community Development, Social Welfare, Gender, Elderly and Children and which distributed albendazole and praziquantel in the context of the programme for morbidity control of schistosomiasis from 2001 to 2007 [9] and, subsequently, during implementation of the Zanzibar Elimination of Schistosomiasis Elimination (ZEST) project during 2012–2017 [10]. More precise data on PC activities conducted between 2011 and 2019 are available and presented in Table 1; during this period, preschool-aged children, school-aged children and adults were targeted. The coverage reported in Table 1 is that reported annually to WHO (calculated using the estimated total number of school-aged children in the area as denominator and the number of children treated as numerator). In 2019 and 2020, no PC activities were conducted due to problems with drug procurement and the interruption of the schistosomiasis programme.

**Table 1. Preventive chemotherapy activities conducted in Pemba and their coverage according to reports from MoH Zanzibar to WHO.**

| Intervention target | Activity | 2011 | 2012 | 2013 | 2014 | 2015 | 2016 | 2017 | 2018 | 2019 | 2020 | 2021 |
|---|---|---|---|---|---|---|---|---|---|---|---|---|
| STH and schistosomiasis | Survey | X | | | | | | | | | | X |
| | Number of rounds | | 2 (Apr–Nov) | 2 (Jun–Nov) | 1 (Apr) | 2 (Jun–Dec) | 2 (May–Nov) | 2 (Mar–Nov) | 2 (Mar–Nov) | 0 | 0 | |
| | Medicine used | | PZQ+ALB | PZQ+ALB | PZQ+ALB | PZQ+ALB | PZQ+ALB | PZQ+ALB | PZQ+ALB | NA | NA | |
| | Population targeted | | preSAC SAC Adults | preSAC SAC Adults | preSAC SAC Adults | preSAC SAC Adults | SAC Adults | SAC Adults | SAC Adults | NA | NA | |
| | Coverage children | | * | * | 82% | 99% | 49% | 64% | 68% | NA | NA | |
| | Coverage adults | | | | 82% | 67% | 47% | 53% | 58% | | | |
| Lymphatic filariasis | Number of rounds | | 0 | 0 | 1 (Nov.) | 0 | 1 (May) | 0 | 1 | 0 | 0 | |
| | Medicine used | | NA | NA | IVR+ALB | NA | IVR+ALB | NA | IVR+ALB | NA | NA | |
| | Population targeted | | NA | NA | SAC Adults | NA | SAC Adults | NA | SAC Adults | NA | NA | |
| | Coverage | | NA | NA | 83% | NA | 84% | NA | 96% | NA | NA | |

ALB = albendazole; IVR = ivermectin; NA = not applicable; PreSAC = preschool-aged children; PZQ = praziquantel; SAC = school-aged children; STH = soil-transmitted helminthiases

* Details about coverage not available.

PC remains the cornerstone for morbidity control of STH infection and its impact should be evaluated periodically. The objective of this study was thus to assess the current distribution of STH in Pemba after more than 25 years of PC considering however that the intervention has not been consistently administered.

## Method

### Ethics statement

Ethical approval for the study was obtained from the Zanzibar Health Research Ethical Committee (ZAHREC), which works under the Zanzibar Health Research Institute (ZAHRI), on October 2020 (REF: No. ZAHREC/03/PR/OCT/2020/24). A formal written consent was obtained from the parents/guardian of the children enrolled in the study.

*Time of survey conduction*: The impact survey was conducted from 1 to 25 March 2021.

*School selection and enrolment of children*: The total number of public primary school in Pemba is 95. For comparison purposes, we selected for the present study the same 24 schools selected in 2011 when, in each district, 6 schools where randomly selected. (Fig 1).

In each selected school, two grades were identified: standard 1 (age interval 6–8 years) and standard 3 (age interval 9–12 years). To ensure comparable numbers and equal participation for both sexes, in each school the children were grouped based on their sex and 70 children were randomly selected in each of the two groups. The study procedure was explained to the selected children, who were given a consent form and a stool container to take home. The specimens collected in the school the following morning were deposited in a cold box and transported to the laboratory for examination in the afternoon.

The specimens were microscopically examined for the presence of STH eggs using the Kato–Katz technique [11]. For each specimen, a single Kato–Katz smear was prepared and the

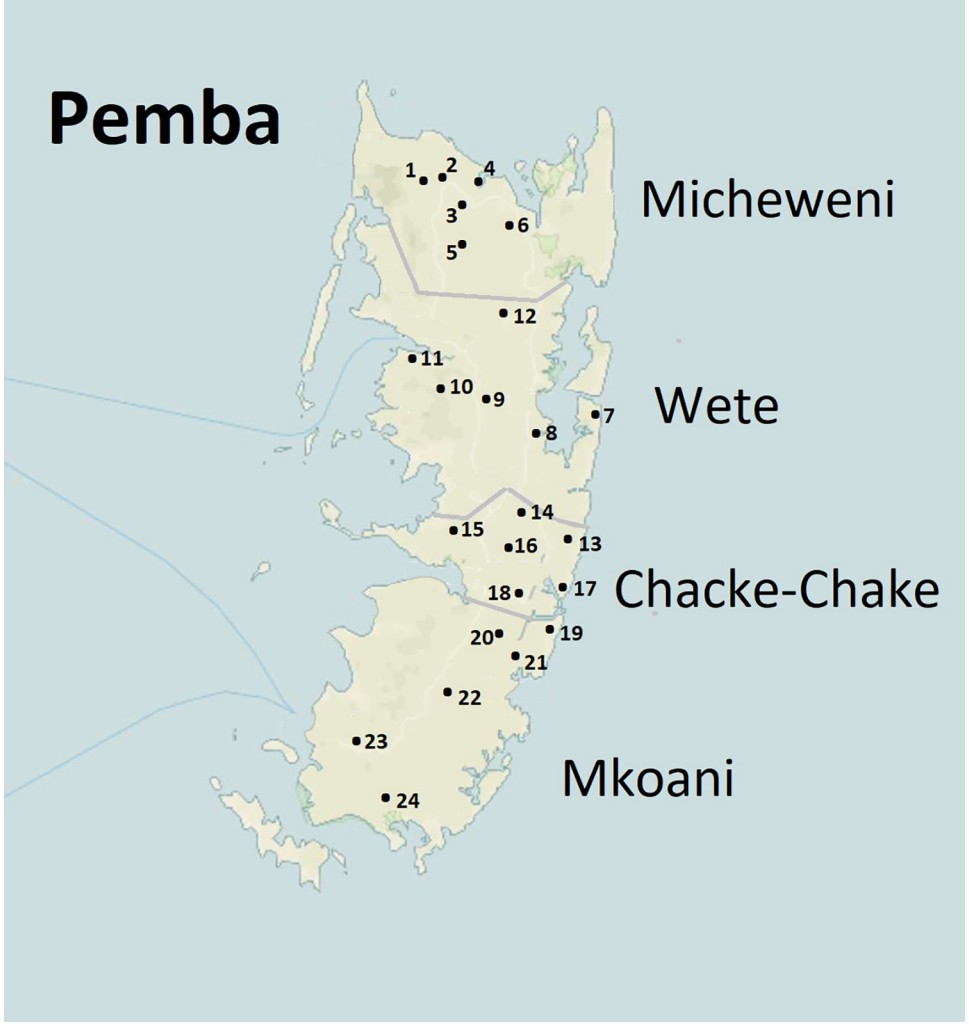

**Fig 1. Location of the 21 primary schools in the sample the number correspond to the school number in Table 2.**
Base map in public domain from U.S. Geological Survey (USGS) via TopoView (https://ngmdb.usgs.gov/topoview/
viewer/#10/-5.2079/39.9923), a tool created by the National Geologic Map Database Project (NGMDB) with a basemap
from OpenStreetMap.

number of eggs detected for each STH species was recorded. For quality control purposes, 10%
of the slides examined every day were randomly selected and read by an experienced laboratory technician.

*Data entry and analysis*: Demographic and parasitology data were entered in EpiInfo version 3.5.1 and later imported to Excel for cleaning. The cleaned data were then imported to Stata 13 (College Station, Texas, USA) for analysis. For each STH species, the number of eggs counted in the slide was then multiplied by a factor of 24 to obtain an egg per gram of faeces. WHO thresholds [3] were used to categorize the intensity of infection as light, moderate or heavy. The Chi Square test ($\chi^2$) was used to assess association of categorical data and the probability of < 0.05 was considered as significant. The student t-test (t-test) was used to estimate group means. The proportion test (z-test) was used to estimate differences in the proportion of infection between different years.

## Results

### Age and sex distribution in the sample

A total of 3454 school children (1749 females and 1705 males) were initially selected, but only the 2474 (71.6%) children who provided stool samples and signed informed consent form were enrolled in the study. Overall, the mean age of enrolled children was of 8.0 years ± 1.8 standard deviation [SD] (95% confidence interval [CI] = 7.9–8.1) with minor difference between boys (8.2 ± 2.0 SD) and girls (7.8 ± 1.7 SD). The mean age of standard 1 children was 6.34 years, and the mean age of standard 3 children was 9.63 years.

### STH prevalence in Pemba

The total prevalence measured was 80% (95% CI 78–81); Of the 2474 children enrolled, 1183 (47.8%) had infection of light intensity and 796 (32.2%) had MHI infection. *T. trichiura* was the more prevalent species (72.3% 95% CI 70–74) and hookworms the less prevalent (18% 95% CI 17–20). Table 2 presents the prevalence of STH infection across schools and districts in Pemba.

Details on the prevalence of the different species and categories of intensity of infections are shown in Fig 2.

### Prevalence of STH by sex and age group

The analysis of STH infection relative to sex showed that boys were slightly more infected than girls (Table 3), but we do not consider this difference significant from a clinical or epidemiological point of view.

The analysis of STH worm infection in relation to school grade levels (Standard 1 to 3) revealed that the prevalence of the different STH species was similar among the two grades.

### Prevalence of STH infection by region, district and school

The prevalence of STH by region, district and school is presented in Table 3.

Micheweni District had the highest STH prevalence (89%); in the other three districts of the island, the STH prevalence was similar (around 76%) ($p < 0.001$).

Michakaini School (Chacke-Chacke District) was the only school with a prevalence of any STH below 50%; seven schools (Chambani, Kinowe, Mkanyageni, S/vyamboni, Tumbe, Wesha and Wingwi A) had a STH prevalence of over 90%.

In all schools, the prevalence of *T. trichiura* was consistently higher (range 43–94%), followed by *A. lumbricoides* (range 22–83%); that of hookworm infections ranged between 0% and 36%.*Mixed infections*

As indicated above, 80% of the children were infected with at least one STH parasite species, 32% were infected with only one species, 33% with two STH species and 15% with three STH species.

### Intensity of infection

The prevalence of the different classes of intensity is presented in Fig 1. The mean intensity of infections is presented in Table 4.

### Trend of STH infection in 1994–2021

In 1994, the STH epidemiological data were collected before any large-scale control intervention; interestingly, only one child of the 3578 investigated tested negative for all three STH species.

**Table 2. Prevalence of STH infection across schools and districts in Pemba.**

| Region | District | N | School | Prevalence STH (%) | | | |
|--------|----------|---|--------|---------|----------------|-----------|--------------|
| | | | | Any STH | A. lumbricoides | Hookworms | T. trichiura |
| **North** | | | | **83.3** | **55.0** | **26.6** | **75.8** |
| | _Micheweni_ | | | _90.0*_ | _69.1*_ | _37.5*_ | _82.7*_ |
| | | 1 | Konde A | 80.6 | 58.3 | 18.5 | 71.4 |
| | | 2 | Konde B | 76.9 | 55.6 | 36.8 | 66.7 |
| | | 3 | Kinowe | 98.3 | 83.8 | 35 | 94 |
| | | 4 | Tumbe | 97.4 | 78.1 | 36 | 91.2 |
| | | 5 | S/vyamboni | 93.5 | 64.8 | 51.9 | 88 |
| | | 6 | Wingwi A | 93.5 | 73.8 | 46.7 | 85.1 |
| | _Wete_ | | | _76.6_ | _40.9_ | _15.7_ | _68.8_ |
| | | 7 | Kangagani | 75 | 35.7 | 9.8 | 70.5 |
| | | 8 | K/minungwini | 89.3 | 61.3 | 14.7 | 81.3 |
| | | 9 | Mzambarauni | 69.4 | 32.4 | 11.1 | 58.3 |
| | | 10 | Piki | 71.4 | 35.2 | 28.6 | 69.2 |
| | | 11 | Bagamoyo | 85 | 57.5 | 16.7 | 67.5 |
| | | 12 | Kizimbani | 69.2 | 23.1 | 13.2 | 65.9 |
| **South** | | | | **75.4** | **43.5** | **10.0** | **67.7** |
| | _Chake-Chake_ | | | _73.1_ | _40.4_ | _7.5_ | _68.1_ |
| | | 13 | Uwandani | 74.1 | 42.6 | 1.9 | 66.7 |
| | | 14 | Ziwani | 70.2 | 33.3 | 15.5 | 64.3 |
| | | 15 | Wesha | 93.8 | 67.1 | 10.4 | 90.6 |
| | | 16 | Michakaini | 49.4 | 20.1 | 0 | 43.2 |
| | | 17 | Pujini | 82 | 48.4 | 8.2 | 79.5 |
| | | 18 | Chanjamjawiri | 68.9 | 31.1 | 8.7 | 64.1 |
| | _Mkoani_ | | | _77.7_ | _46.5_ | _12.5_ | _67.4_ |
| | | 19 | Chambani | 90.3 | 67 | 11.7 | 75.7 |
| | | 20 | Ngwachani | 71 | 38 | 7 | 52 |
| | | 21 | Mizingani | 85 | 48 | 18 | 76 |
| | | 22 | Mtambile | 70 | 38 | 8 | 67 |
| | | 23 | Ng'ombeni | 56.8 | 22.1 | 4.2 | 50.5 |
| | | 24 | Mkanyageni | 93 | 65.8 | 26.3 | 83.3 |

* Significantly different from other districts (p < 0.001).

Fifteen years of PC interventions (1995–2011), even if not consistently applied, reduced the prevalence of STH in schoolchildren from 100% to 88% and the prevalence of MHI infections from 76% to 29%. Additional PC interventions conducted between 2011 and 2021 further reduced the prevalence of STH to 79%. In parallel, the prevalence of MHI infection fell from 76% in 1994 to 32% in 2021. The trend of STH prevalence by STH species and the prevalence of MHI infections are presented in Fig 2.

In terms of STH species, the main prevalent species in 1994 were _T. trichiura_ and hookworms, with prevalences of 96% and 94% respectively. In 2021 the prevalence of _T. trichiura_ remains high at 73%, whereas that of hookworm has fallen to 19%.

Regarding reductions in intensity of infections of MHI, hookworm infections reduced consistently from 19% in 1994 to 1% 2021, _T. trichiura_ infections from 41% in 1994 to 18% in 2021 and _A. lumbricoides_ infections increased slightly from 20% in 1994 to 24% in 2021 (Fig 2).

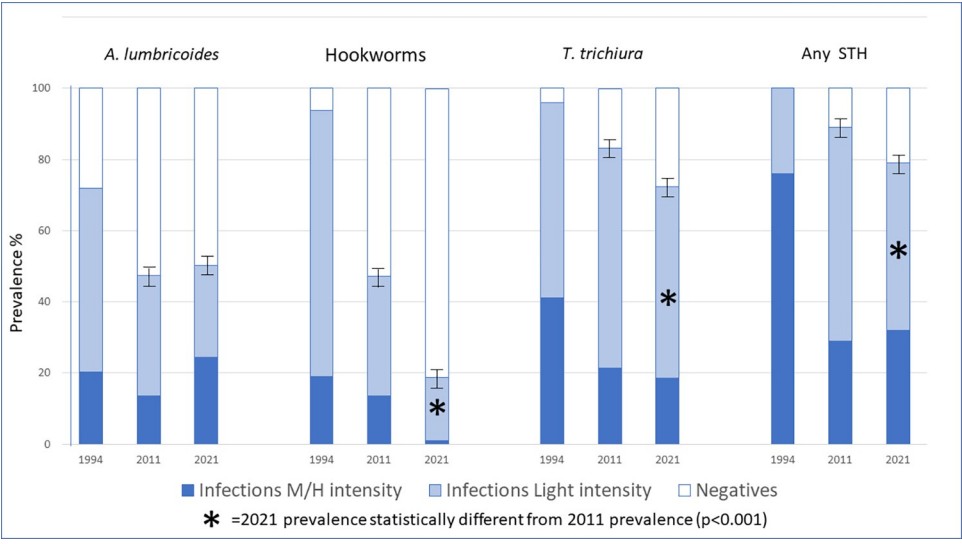

**Fig 2. Categories of intensity of infection for each of the three STH parasites and for any infection from 1994 to 2021.**

## Discussion

PC is advocated by WHO and is considered the cornerstone for morbidity control and, possibly, elimination of many neglected tropical diseases including STH infection [12]. In Pemba, the results of a PC intervention conducted for 25 years are not optimal: while the prevalence of hookworms has been significantly reduced and infections of MHI with this parasite almost eliminated, the prevalence and intensity of *A. lumbricoides* and, particularly, *T. trichiura* remain high.

These results differ from those reported by Mupfasoni and colleagues [12], who examined the result of impact surveys in 15 countries in Africa, America and Asia, which show that, for all species a prevalence reduction of more than two-thirds is normally obtained after 5 years of PC intervention.

The differences between our study and the results reported by Mupfasoni and colleagues are probably linked to the extremely high STH transmission in Pemba: prevalence near 100%, as observed in Pemba in 1994, was extremely rare even in the absence of PC interventions [13–16]. In our opinion, the consistently high levels of prevalence and the intensity of infection in Pemba result from several factors: (i) the favourable climatic conditions for STH transmission (in terms of vegetation index, rainfall, soil pH, soil moisture and mean temperature) [17]; (ii) the population density (387 persons/km$^2$), which is among the highest in sub-Saharan Africa [18]; and (iii) the very poor sanitation coverage, especially in rural areas like Pemba (27% of households in Zanzibar use unimproved toilet facilities or have no toilet facilities at all) [19].

**Table 3. Proportions of boys and girls infected with different species of STH.**

| | Parasite species | | | |
|---|---|---|---|---|
| | *A. lumbricoides* | Hookworm | *T. trichiura* | Any STH |
| Girls | 45.2%* | 15.4%* | 71.9% | 78.5% |
| Boys | 55.6%* | 22.5%* | 72.8% | 81.5% |

* Difference between girls and boys significantly different with p < 0.01.

**Table 4. Mean intensity of infection in eggs per gram by species.**

| STH species | Survey 2011 | Survey 2021 |
|---|---|---|
| *A. lumbricoides* | 2 279 (CI 2205–2502) | 5979 (CI 5445–6512) |
| *T. trichiura* | 716 (CI 664–767) | 702 (CI 600–805) |
| Hookworms | 247 (CI 225–268) | 105 (CI 89–121) |

An additional possibility is the development by the parasites (particularly *T. trichiura* which is the predominant species) of resistance against albendazole as speculated previously [20–21]. On the one hand, further studies to elucidate the existence of mutations in the STH genome leading to resistance to benzimidazole derivatives are under way. On the other hand, a recent study from Keiser and collaborators [22] found that the egg reduction rate obtained with albendazole for trichuriasis in Pemba was of 57%, which is within the range of efficacy indicated by WHO as normal [23].

The observation of such high prevalence of STH infection among school-aged children also raises concerns about reported drug coverage and children's compliance with treatment, as these two indicators play a critical role in achieving the objectives of PC interventions [24].

Another interesting aspect of this study is the different performance of the STH species over time: *A. lumbricoides* shows an important reduction in prevalence but a paradoxical increase in infection intensity, whereas hookworms and *T. trichiura* show a sustained decrease in prevalence and intensity. These results are surprising because the efficacy of albendazole and mebendazole is known to be better on *A. lumbricoides* than on hookworms [20] and the fact that in 1994 the prevalence of hookworms was higher than that of *A. lumbricoides* [6].

A possible explanation is the different transmission pattern of the STH parasites that entails for hookworms the development of larval stages and skin penetration (the large majority of hookworms in Pemba is *N. americanus* [25]). As consequence, the free-living stages of this parasite in the soil are probably more vulnerable than those of *A. lumbricoides* and *T. trichiura* eggs, which can survive in the environment for several years [26]. In this situation, a lower impact of the anthelminthic on the parasite while in the host may produce a larger negative impact on the transmission capacity of the parasite. This is confirmed by the review of Mupfasoni et al. [12] in which in 75% of the countries examined the reduction after over 5 years of PC was higher for hookworm (average 70%) than for *A. lumbricoides* (average 57%) and *T. trichiura* (average 47%).

The study conducted in Pemba has some limitations. First, the record of the interventions implemented before 2012 are incomplete, coverage of the groups at risk is also not always optimal, and the survey was conducted more than 3 years after the last PC intervention.

In conclusion, PC interventions were implemented for over 25 years in Pemba yet, despite their beneficial impact on prevalence and intensity of STH infections, the interventions did not sufficiently control STH morbidity. This is probably due to a combination of multiple factors: irregular PC, climatic conditions extremely favourable to STH transmission, the low sensitivity of *T. trichiura* to benzimidazoles, high population density and poor sanitation.

We consider that, when the STH baseline prevalence is extremely high, as in Pemba, it will be extremely difficult to reach the objectives set by WHO for 2030, [27] if PC intervention is not accompanied by substantial improvement of sanitation.

In 2019, Zanzibar launched a 10-year comprehensive cholera elimination plan, which included diagnosis and case management, vaccination, a surveillance system, health promotion activities, and plans to improve infrastructure for the provision of safe water and sanitation facilities. We hope that the sanitation component of this strategy, complementing PC efforts, will reduce STH transmission in the island. Possible changes in PC approaches to

better control STH in Pemba would be (i) to assure high coverage and compliance in all schools (ii) to use mebendazole instead of albendazole because it has better activity on *T. trichiura* [28], which currently appears to be more problematic than hookworm, and (iii) to co-administer ivermectin with mebendazole to further increase drug efficacy [28].

We did not consider the extension of PC to the entire population because the higher cost of community distribution is estimated to be 10 times higher than that for distribution through the school system for school-aged children or during child health days for preschool-aged children [29] and because of the risk to stimulating drug resistance in the parasite if PC is administered for a long time to the entire population without refugia for the parasite [30].

## Disclaimer

The authors alone are responsible for the views expressed in this article and they do not necessarily represent the views, decisions or policies of the institutions with which they are affiliated.

## Supporting information

**S1 Survey 2011. Dataset of 2011 survey.**
(XLSX)

**S2 Survey 2021. Dataset of 2021 survey.**
(XLSX)

## Author Contributions

**Conceptualization:** Shaali Ame, Pauline Mwinzi, Antonio Montresor.

**Data curation:** Shaali Ame.

**Formal analysis:** Shaali Ame, Antonio Montresor.

**Investigation:** Shaali Ame.

**Methodology:** Shaali Ame.

**Project administration:** Shaali Ame, Said Mohammed Ali.

**Supervision:** Pauline Mwinzi, Said Mohammed Ali.

**Validation:** Said Mohammed Ali.

**Visualization:** Antonio Montresor.

**Writing – original draft:** Antonio Montresor.

**Writing – review & editing:** Fatma Kabole, Alphoncina Masako Nanai, Denise Mupfasoni, Said Mohammed Ali.

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
