## [Decision Letter · Decision Letter 0]

8 Feb 2022

Dear Dr. Montresor,

Thank you very much for submitting your manuscript "Impact of preventive chemotherapy on transmission of soil-transmitted helminth infections in Pemba Island, United Republic of Tanzania, 2011–2021" for consideration at PLOS Neglected Tropical Diseases. As with all papers reviewed by the journal, your manuscript was reviewed by members of the editorial board and by several independent reviewers. In light of the reviews (below this email), we would like to invite the resubmission of a significantly-revised version that takes into account the reviewers' comments. 

In addition to the reviewers comments please add CI to the text and figures (to all values of % throughout the manuscript). 

When comparing prevalence there should be mention of the p value and CI - only with this information the authors can make statements regarding whether the prevalence reduced or increased - comparing just the % as currently is done is not appropriate.

Also, given the comments of rev. 2, please review your manuscript following the STROBE guidelines and include a filled in form as a Supporting Information file. This will ensure that you provide more details and may help with more careful consideration of potential biases. 

We cannot make any decision about publication until we have seen the revised manuscript and your response to the reviewers' comments. Your revised manuscript is also likely to be sent to reviewers for further evaluation.

Sincerely,

Susana Vaz Nery

Associate Editor

Hélène Carabin

Deputy Editor

In addition to the reviewers comments please add CI to the text and figures (to all values of % throughout the manuscript). 

When comparing prevalence there should be mention of the p value and CI - only with this information the authors can make statements regarding whether the prevalence reduced or increased - comparing just the % as currently is done is not appropriate.

Reviewer's Responses to Questions

**Key Review Criteria Required for Acceptance?**

**Methods**

-Are the objectives of the study clearly articulated with a clear testable hypothesis stated?

-Is the study design appropriate to address the stated objectives?

-Is the population clearly described and appropriate for the hypothesis being tested?

-Is the sample size sufficient to ensure adequate power to address the hypothesis being tested?

-Were correct statistical analysis used to support conclusions?

-Are there concerns about ethical or regulatory requirements being met?

Reviewer #1: Methods: 

The objective of the study is clear. However, since the authors declare that the records of PC from 1994 to 2011 are imprecise it does not seem completely correct to speak about “more than 25 five years of PC”. The statement gives the idea of continuous and consistent PC campaigns during that period which is not the real scenario of the study. Being Pemba a high prevalence area where reinfection is very possible and PC should be implemented twice a year. It is not fear for the PC strategy to evaluate its effectiveness in a setting where: i) the coverage of the PC is imprecise (for example 2016: 47 – 90%) and ii) two years have passed since the last PC round was administered (2019 and 2020). 

The design is appropriate for the objective. The diagnostic method used (Kato-Kats) is the recommended by WHO for diagnosis and monitoring of STH. An adequate quality control was also performed.

A definition and description of the formula used to calculate coverage is needed, as well as clarification about the source of the data used for that indicator. 

More information is needed about population. What ages have the children of grades standard 2 and standard 3? Not all the PC interventions targeted preSAC children. Then children younger than 7 years might have never received PC. The inclusion of those children in the study would underestimate the impact of the PC. 

The sample size is sufficient; actually the number of children analyzed is impressive. 

No ethical concerns arise from reading of the study.

Reviewer #2: 1. One of my main concerns is about the content of methods section. While the methods used were appropriate for this survey, the content of methods section is too brief. More important information should be provided. For instance, description of the study area, study population, epidemiological characteristics of the targeted population pertaining to intestinal parasites transmission, students’ participation, samples delivery and transport to examination station, etc. 

2. Also, provide information on the selection of schools and the procedures of sampling. It is not enough to just refer the reader to the 2011-study (2018 PhD thesis).

3. Please state the time frame of the survey. When was this study conducted?

Reviewer #3: See below

**Results**

-Does the analysis presented match the analysis plan?

-Are the results clearly and completely presented?

-Are the figures (Tables, Images) of sufficient quality for clarity?

Reviewer #1: The results match the analysis plan and are completely presented. 

It would be interesting to know which the hookworm species more prevalent in the island is. Obviously that could not be known from the Kato-kats results. But maybe that information is known from previous studies or from other laboratory techniques used for control purposes. 

Figures 4 resume clearly the results. Figures 1, 2 and 3 seem repetitive. A map of the island showing the information presented in table 3 would be clarifying and eye-catching.

Reviewer #2: 1. Line 131: “495 (20%) where not infected,” this can be removed. Please indicate the overall prevalence of STH.

2. Line 142-143: Add the statistics. Was the difference significant or not? What species?

3. Prevalence of STH by sex: Provide the actual p values where needed. A row can be added to table 2 and p values for all parasite groups can be given.

4. Table 3: add rows for total for each district and report statistical tests.

5. Table 3: report statistical tests for difference between both regions, i.e. north and south.

6. Prevalence of STH infection by age: What about difference according to STH species? Please report this and provide statistics.

7. Prevalence of STH infection by age: this subsection can be merged with the “Prevalence of STH by sex”. Results according to age can be added to table 2.

8. Line 197: “23% in 1994 to 24% in 2021” but this is not a reduction! Please revise these percentages for Ascaris as they are incompatible with Figure 4.

9. It will be useful to display the prevalence % over the columns of all figures.

Reviewer #3: See below

**Conclusions**

-Are the conclusions supported by the data presented?

-Are the limitations of analysis clearly described?

-Do the authors discuss how these data can be helpful to advance our understanding of the topic under study?

-Is public health relevance addressed?

Reviewer #1: The conclusions are supported by the data presented except for the statement: “the interventions did not sufficiently control STH morbidity” (lines 234 and 235). None of the results presented actually measure morbidity (anemia, malnutrition, stunting). Even though it is reasonable to deduce that the morbidity probably remains high with the very high prevalence and infection intensity found. 

The limitations of the analysis are not clearly described. The limitations regarding coverage and records of the PC interventions should be discussed before stating: “In Pemba, the results of a PC intervention conducted for 25 years are not optimal”. Instead it could be more realistic to conclude that: The results of the PC are not optimal with the poor coverage achieved and targeting preSAC in only 4 of the 10 interventions recorded. 

The results show a very interesting difference in response of each STH species to the PC. Trichuris trichiura remains with high prevalence but MHI proportion is reduced. Ascaris lumbricoides shows an important reduction in prevalence but a paradoxical increase in infection intensity. Hookworms show a sustained decrease in prevalence and intensity. These differences should be more fully discussed in the paper for two reasons: 1) they have morbidity implications, and 2) raise the need to establish individual transmission models for each STH species and not seek a single explanation for all. Particularly the findings on hookworms are interesting and require a possible explanation. The efficacy of albendazole is not higher against hookworms than against Ascaris lumbricoides. The reason for the different response of these two species might be an improvement in sanitation facilities for example. 

The authors present a possible strategy to improve the results of PC in Pemba based on their results. The option of using mebendazole instead of albendazole should be more fully discussed presenting its advantages and disadvantages. With this strategy some little efficacy against Trichuris trichiura might be gained but a lot of efficacy against hookworms would be lost. The strategy of consistently targeting preSAC and adults in the PC interventions should be also discussed. 

The relevance of the study for public health is addressed.

Reviewer #2: 1. Please describe the limitations of the study. For example, using a single faecal sample and single Kato-Katz smear, study design, etc.

Reviewer #3: see below

**Editorial and Data Presentation Modifications?**

Reviewer #1: (No Response)

Reviewer #2: 1. The abstract in general is suitable for authors summary required by PLoS NTD; and another structured abstract can be provided. In the abstract, results should be provided clearly with proportions and statistics.

2. Abstract, first sentence: Why only focused on the STH impact on nutritional status? A more general statement can be used.

3. Line 39: “periodic treatment” can be replaced by periodic preventive chemotherapy or MDA.

4. Lines 41-43: This statement about the 2 surveys can be rewritten in a more general statement to indicate the continued burden over time. The sentence is repeated in the 3rd paragraph of introduction section.

5. Line 49: “… infections are higher, ..” or remin high!

6. Lines 50-52: Did you investigate these factors? This statement is suitable for authors summary but not for the main abstract section.

7. Lines 57-58: ” points ii & iii” did you investigate these points among the studied population?

8. Lines 60-61: Theoretically, Strongyloides stercoralis is also a STH species.

9. Table 1: correct “toto” in the title.

10. Table 1: Why PC activities were stopped in 2019 & 2020? Please indicate this in last paragraph of introduction, if applicable.

11. Table 1: Correct the “Number of rounds” for lymphatic filariasis in 2018. It cannot be 0!

12. Line 117: Correct “of ≤ 0.05” to “of < 0.05”.

13. Line 132: remove “only” and “at least”.

14. Line 220: considering point #iii, it is unclear whether the targeted schools and populations are rural or urban.

15. The results showed a small increase in the prevalence of MHI in 2021 compared to 2011 (Fig 2) and it seems to be due to Ascaris (Fig 4). This point can be discussed. Was it due to ceased PC activities in 2019-2020?

Reviewer #3: See below

**Summary and General Comments**

Reviewer #1: This is a very interesting study. Relevant for the area of helminthology and also for public health. It has two major strengths: 1) the long period of follow up and 2) the big sample size. Its weaknesses are: 1) the lack of a global picture since preSAC and adults (especially woman of childbearing age) were not studied and 2) the long period between the last PC intervention and the surveillance study.

Reviewer #2: This manuscript describes a large-scale survey that aimed to determine the prevalence of STH infections among schoolchildren in 4 districts of Pemba Island with the ultimate objective was to assess the impact of more than 25 years of PC on the burden of STH in Pemba. Overall, the study reported beneficial impact of the PC intervention; however, the impact was not optimal due to the lack of other interventions essential to sustain the impact of PC in any endemic area.

Overall, the topic is interesting and important. The findings shed light on the long-term impact of PC intervention in highly endemic areas and hence, worth publication. However, the manuscript has some drawbacks that should be addressed in a revised version.

Reviewer #3: General: Shaali Ame and co-authors report the results of a third soil-transmitted helminth (STH) survey among school children in Pemba, this time 25 years after initiation of preventive chemotherapy (PC) in the island. They found a very high Trichuris trichiura prevalence and high Ascaris lumbricoides prevalence while the hookworm prevalence decreased considerably. Various explanations for this persistence of high STH prevalences are discussed. This is a refreshingly short and clear manuscript, and only few comments are offered for consideration:

- Abstract and discussion: while climatic conditions are certainly ideal for STH this probably is not the key factor since they are equally good in many places with lower (albeit often still high) STH prevalences

- Abstract and discussion: high T. trichiura transmission is cited as an explanation for the consistently high STH prevalence. However, the transmission obviously is the outcome, not an explanation…

- Please state the selection criteria for the surveyed schools in 2011. How representative are they for the island?

- Is anything known about treatment coverage in the survey schools and communities? Overall coverage seems to be quite high, though…

- Have treatment coverage figures been verified (e.g. with post-treatment surveys)?

- In addition to the infection intensity classes it would be interesting to also add mean/median egg count figures (and compare them with 2011 values, if possible). 

- Not all statistical test results seems to have been reported. E.g. the statistics of between-survey differences are not evident. 

- No data is presented on STH prevalences among adults. Most likely, it is also high. In combination with a relatively high proportion of households without or with only unimproved sanitation, they are likely to contribute substantially to transmission. Hence, expanding deworming also to adults might be considered and should at least be discussed among possible options to reduce transmission in the short term. 

- Carefully edit the text to eliminate typos, e.g. line 131 “where” should be “were”

PLOS authors have the option to publish the peer review history of their article (what does this mean?). If published, this will include your full peer review and any attached files.

Reviewer #1: No

Reviewer #2: Yes: Hesham M. Al-Mekhlafi

Reviewer #3: Yes: Peter Steinmann
---

## [Decision Letter · Decision Letter 1]

9 May 2022

Dear Dr. Montresor,

We are pleased to inform you that your manuscript 'Impact of preventive chemotherapy on transmission of soil-transmitted helminth infections in Pemba Island, United Republic of Tanzania, 1994–2021' has been provisionally accepted for publication in PLOS Neglected Tropical Diseases.

Best regards,

Susana Vaz Nery

Associate Editor

Hélène Carabin

Deputy Editor

Reviewer's Responses to Questions

**Key Review Criteria Required for Acceptance?**

**Methods**

-Are the objectives of the study clearly articulated with a clear testable hypothesis stated?

-Is the study design appropriate to address the stated objectives?

-Is the population clearly described and appropriate for the hypothesis being tested?

-Is the sample size sufficient to ensure adequate power to address the hypothesis being tested?

-Were correct statistical analysis used to support conclusions?

-Are there concerns about ethical or regulatory requirements being met?

Reviewer #1: The objective of the study is clear.

The design is appropriate for the objective. The diagnostic method used (Kato-Kats) is the recommended by WHO for diagnosis and monitoring of STH. An adequate quality control was also performed.

A definition and description of the formula used to calculate coverage was given, as well as clarification about the source of the data used for that indicator.

The study population is sufficently described, details on the age of the chilfren enrolled is given.

The sample size is sufficient; actually the number of children analyzed is impressive.

No ethical concerns arise from reading of the study.

Reviewer #2: All are ok.

Reviewer #3: see below

**Results**

-Does the analysis presented match the analysis plan?

-Are the results clearly and completely presented?

-Are the figures (Tables, Images) of sufficient quality for clarity?

Reviewer #1: The analysis presented match the analysis plan.

The results are clearly and completely presented. More important details about the findings of the surveillance were given in comparison to the previous version of the manuscript. For example the Table 4 showing the mean EPG of each STH species in 2011 and 2021 surveys is very illustrative. The map enhances the results presentation.

The images and tables are sufficient for clarity.

Reviewer #2: All are ok.

Reviewer #3: see below

**Conclusions**

-Are the conclusions supported by the data presented?

-Are the limitations of analysis clearly described?

-Do the authors discuss how these data can be helpful to advance our understanding of the topic under study?

-Is public health relevance addressed?

Reviewer #1: The conclusions are supported by the data presented, except for the statement: “the interventions did not sufficiently control STH morbidity” (lines 265 and 267) since none of the results presented actually measure morbidity (anemia, malnutrition, stunting).

The conclusions are considerably improved in comparison to the previous version of the article. The limitations are fully and clearly described. An interesting explanation is presented for the differences found among STH species after regular PC campaigns.

The discussion about how the findings of this study can be helpful for Pemba island, and also for global health is one of the strengths of the study.

Public health relevance is addressed and possibility to generalize the findings is also discussed.

Reviewer #2: All are ok.

Reviewer #3: see below

**Editorial and Data Presentation Modifications?**

Reviewer #1: (No Response)

Reviewer #2: None

Reviewer #3: see below

**Summary and General Comments**

Reviewer #1: This is a very interesting study. Relevant for the area of helminthology and also for public health. Addresses some key points of the WHO preventive chemotherapy recommendations that are under revision (the target group, the recommended coverage and drugs, etc). Manuscripts from different “real world” experiences with PC such as this enhance the discussion about how to improve current guidelines to accomplish STH control.

Reviewer #2: The authors have properly addressed my comments and I am satisfied with the way they have improved the manuscript.

Reviewer #3: The authors have adequately responded to the peer-review comments and the amended manuscript reflects their consideration of the comments.

Two minor points:

- line 153: the word "of" after the semicolon should not be capitalized

- line 298: delete "higher"

line numbers refer to the manuscript with track-changes

PLOS authors have the option to publish the peer review history of their article (what does this mean?). If published, this will include your full peer review and any attached files.

Reviewer #1: No

Reviewer #2: **Yes: **Hesham M. Al-Mekhlafi

Reviewer #3: No

---

## [Editor Report · Acceptance letter]

10 Jun 2022

Dear Dr. Montresor,

We are delighted to inform you that your manuscript, "Impact of preventive chemotherapy on transmission of soil-transmitted helminth infections in Pemba Island, United Republic of Tanzania, 1994–2021," has been formally accepted for publication in PLOS Neglected Tropical Diseases.

Best regards,

Shaden Kamhawi

co-Editor-in-Chief

Paul Brindley

co-Editor-in-Chief
